# Analysis of a Library of *Escherichia coli* Transporter Knockout Strains to Identify Transport Pathways of Antibiotics

**DOI:** 10.3390/antibiotics11081129

**Published:** 2022-08-19

**Authors:** Lachlan Jake Munro, Douglas B. Kell

**Affiliations:** 1Novo Nordisk Foundation Center for Biosustainability, Technical University of Denmark, 2800 Lyngby, Denmark; 2Institute of Systems, Molecular and Integrative Biology, University of Liverpool, Liverpool L69 3BX, UK

**Keywords:** transporters, antibiotics, *Escherichia coli*

## Abstract

Antibiotic resistance is a major global healthcare issue. Antibiotic compounds cross the bacterial cell membrane via membrane transporters, and a major mechanism of antibiotic resistance is through modification of the membrane transporters to increase the efflux or reduce the influx of antibiotics. Targeting these transporters is a potential avenue to combat antibiotic resistance. In this study, we used an automated screening pipeline to evaluate the growth of a library of 447 *Escherichia coli* transporter knockout strains exposed to sub-inhibitory concentrations of 18 diverse antimicrobials. We found numerous knockout strains that showed more resistant or sensitive phenotypes to specific antimicrobials, suggestive of transport pathways. We highlight several specific drug-transporter interactions that we identified and provide the full dataset, which will be a useful resource in further research on antimicrobial transport pathways. Overall, we determined that transporters are involved in modulating the efficacy of almost all the antimicrobial compounds tested and can, thus, play a major role in the development of antimicrobial resistance.

## 1. Introduction

Antibiotic resistance is a major global healthcare burden, with over 1 million deaths attributable to antibiotic resistance in 2019, and with the World Health Organization listing antimicrobial resistance as one of the top 10 global public health threats facing humanity [1,2]. Despite a pressing need for novel antibiotics that can overcome resistance, few novel drugs have been developed and approved in the last 20 years, with a general downward trend in new approvals since 1983 [3], albeit the last decade has seen some increase in new approvals [4]. While systems have been proposed to overcome the economic obstacles to incentivize antibiotic development [5], other solutions are needed.

In order to exert their effects, most antibiotics must first cross the bacterial cell membrane. There is now strong evidence to suggest that in order to enter cells, nearly all compounds must pass through membrane transporters and that “passive bilayer diffusion is negligible” [6,7,8]. The importance of transporters is further highlighted by the well-established fact that a major mechanism of antibiotic resistance is through adaptations in transport systems. These adaptations reduce the intracellular concentration of antibiotics through either the increased efflux or the decreased influx of antibiotics through membrane transporters [9,10,11]. Antibiotic adjuncts that inhibit the efflux of the active drug have been proposed as a means to counteract resistance [12,13].

While modelling suggests that mutations that cause antibiotic resistance will predominantly be in exporters rather than importers [14], there are also examples where adaptations that reduce the import of antibiotics confer resistance. This has been observed in the case of fosfomycin [15], aminoglycosides [16] and chloramphenicol [17]. 

Of the ~4401 genes in the *Escherichia coli* K-12 chromosome, an estimated 598 encode established or predicted membrane transporters [18]. Despite extensive research, around a quarter of these transporters are orphans (also known as y-genes), in that they have no assigned substrate. Recent studies have illustrated that accumulation of cationic fluorophores involves multiple influx and efflux transporters, with a library of transporter knockouts spanning a 30–70-fold difference in fluorescence level when exposed to the fluorophores SYBR Green or diS-C3(5) [19]. We hypothesized that accumulation and excretion of antimicrobial agents may similarly involve numerous membrane transporters. Previously, we have used a high-throughput screening method to identify a melatonin exporter [20]. In this, growth of a library of knockout transporters was measured in the presence of melatonin, and inhibited growth was used as a proxy to identify when an exporter had been knocked out (conversely, a resistant strain potentially has an importer knocked out).

In the present study, which forms part of an ongoing project to deorphanize all orphan *E. coli* transporters, we investigated the previously developed library of 447 *E. coli* transporter knockouts [19,20] for growth, in the presence of 18 structurally diverse antimicrobial agents. We included novel and recently approved antibiotics, such as cefiderocol and flumequine, as well as drugs not traditionally used against Gram-negative bacteria. We also developed an automated workflow for the liquid-handing steps and data processing. The full dataset encompasses approximately 8000 growth curves (in duplicate), and the entire raw dataset and code used for processing are provided in the Appendix A. We identified several transporter knockout strains that showed resistance or sensitivity to many of the antimicrobials tested, including previously unannotated transporters or y-genes. Finally, we illustrate the potential utility of large-scale analysis of the dataset for predicting substrates of orphan transporters.

## 2. Results

### 2.1. Antimicrobial Selection

We selected a range of compounds that were available and readily soluble in our LB media. We endeavored to include compounds from several major antibiotic classes as well as compounds with activity that have previously not been tested extensively against Gram-negative bacteria (ornidazole and paraquat). Cefiderocol was very recently approved, so was included due to novelty. 

### 2.2. Determination of Antibiotic Concentrations for High-Throughput Screening

The workflow for the screening of each compound (Appendix A) initially required identifying a sub-inhibitory concentration of the antibiotic in the wild-type strain BW1556 (WT, parent strain for the Keio collection). Minimum inhibitory concentrations (MICs) were determined by measurement of *E. coli* growth in a two-fold serial dilution of the relevant antibiotic in LB. We defined MIC values as those with OD levels at 48 h less than 10% of the antibiotic-free media condition. For screening of the transporter library, we selected concentrations that showed some inhibition of growth in the WT strain without causing full inhibition. Generally, this was a concentration half that of the MIC; however, when effects on growth (reduction in maximum OD, increased lag time, decreased growth rate, or some combination thereof) could be clearly seen at concentrations significantly lower than that, then these concentrations were used. The 18 antimicrobial compounds included in this study, as well as the MIC and screening concentrations, are listed in Table 1, and chemical structures of the antibiotics used are shown in Appendix A. 

### 2.3. Baseline Characteristics of Transporter Library

Generally, the growth of nearly all transporter knockout strains in LB without supplementation was similar. The empirical area under the curve (AUC) (a general measure of growth) had a mean of 30.0 and an interquartile range (IQR) of 1.90, while the maximum growth rate had a mean of 1.19 and an IQR of 0.14 h^−1^. The WT growth rate was slightly higher than the mean growth rate, possibly due to the burden of expression of the constitutively active kanamycin cassette present in all knockout strains [21,22]. Histograms illustrating the distribution of the AUC and growth rate are shown in Figure 1A,B. We also observed good correlation between replicates with regards to growth rate and AUC (Figure 1C,D). It should be noted that the automated inoculation step occasionally did not dispense a droplet into the well. In cases where this was clearly the case (i.e., where we observed growth in one but not another replicate), the replicate missing growth was filtered from the results. 

Compared to growth in LB, we saw a much larger degree of variation in growth in the presence of the antimicrobial compounds, consistent with multiple transporter knockouts influencing the level of intracellular antibiotic accumulation. Figure 2A shows rank-order curves for the mean AUC for growth of the transporter library in LB, while Figure 2B–D shows the same information for growth in the indicated antibiotics. In the presence of antibiotics, the rank-order curves show a much wider spread of values. Figure 3 shows a box and whisker plot illustrating the distribution of AUC values for all compounds tested.

### 2.4. Resistant and Sensitive Transporter Results

We observed that many transporter knockouts had effects on the sensitivity to the compounds tested. Due to the large volume of data generated, detailed descriptions of all relevant results are beyond the scope of this paper. As we intend to perform targeted follow-up validation experiments, given that this is an initial exploratory study, we are hesitant to define specific criteria for sensitivity or resistance. Indeed, one advantage of our study over previous high-throughput growth assays is the generation of full growth curves rather than the reduction in growth of a single metric, as we elaborate on in the discussion.

Selected specific results are discussed below. Generally, when searching for novel results, we looked at strains ranked highest or lowest in normalized growth rate or AUC and then manually inspected the growth curves. We selected those highlighted in this paper based on interactions with y-genes or where we felt there were plausible and interesting mechanisms for discussion. 

Our study replicated some previously established drug–transporter interactions. First, we observed that knockout of acrB, a promiscuous drug-efflux protein well-known to play a role in export of many xenobiotics [23], caused increased sensitivity to 10 out of the 18 antimicrobial compounds tested. Interestingly we saw the increased inhibition caused by knockout of acrB take different forms: growth rate decreases including increased lag time, decreased max OD and AUC and complete inhibition (Figure 4). Fosphomycin has been shown to gain access to the cell via the hexose-6-phosphate:phosphate antiporter uhpT and the glutamate/aspartate: H+ symporter gltP [15,24,25]. In our phosphomycin experiment, knockouts of each of these transporters were among the most resistant of the transporter knockout strains tested, consistent with these knockouts causing reduction in influx (Appendix A). MacAB-Tolc has been reported to efflux macrolide antibiotics, and expression of this complex in a strain hypersensitive to antibiotics conferred resistance to macrolides including azithromycin [26]. In our study, the macB knockout strain (∆*macB*) did not result in azithromycin sensitivity, and we also could not see evidence of increased sensitivity to azithromycin in a previously published dataset [27].

Ornidazole is a nitroimidazole generally used to treat protozoan infections, and, in this study, we found it to have an MIC in *E. coli* BW1556 of 800 mg/L (Table 1). The transporter nimT (previously yeaN) has been shown to function as an exporter of 2-nitroimidazole, a compound structurally similar to ornidazole [28]. We saw only a modest increase in ornidazole sensitivity in ∆*nimT* compared to WT (Figure 5A). We did, however, observe a clear increased sensitivity to ornidazole in ∆*argO* (Figure 5B), a strain with a probable arginine exporter knocked out [29]. Some strains also showed an ornidazole-resistant phenotype. These included ∆*narU*, a nitrite/nitrate exchanger knockout (Figure 5C). Possibly, this transporter is interacting in some way with the nitro-group on ornidazole, and the knockout of this reduces influx into the cell. We also saw a resistant phenotype in ∆*ygaH* (Figure 5D), an uncharacterized orphan transporter. The other most resistant and most sensitive strains (by normalized AUC) are shown in Table 2.

Azithromycin is a macrolide antibiotic, widely used in the treatment of clinical infections. We saw several transporter knockout strains that had dramatically decreased growth in the presence of 7.5 mg/L azithromycin compared to WT. These included the choline transporter knockout ∆*betT* and a tyrosine symporter knockout ∆*tyrP* (Figure 6A,B). Near-complete inhibition was also seen in the multidrug exporter knockout strain ∆*mdtN*. We also observed resistant phenotypes, specifically in the orphan knockout strains ∆*yhdW* and ∆*ydfJ* (Figure 6C,D). Growth parameters for the most resistant and sensitive strains are shown in Table 3.

We also included compounds not traditionally used as antibiotics, which have antimicrobial activity. Paraquat (also known as methyl viologen) is a widely used herbicide, which exerts toxic effects, after conversion to a superoxide radical, once inside the cell [30]. We found that concentrations up to 257 mg/L did not cause full inhibition of *E. coli* growth; however, there was evidence of growth inhibition at concentrations of 32 mg/L (Figure 7A). We screened the growth of the transporter knockout library at 67 mg/L and observed several strains that had increased sensitivity including the y-gene knockout ∆*ydcZ* and in two metal cation exporter knockouts ∆*cusA* (Figure 7B) and ∆*fieF*, suggestive of the ability of these transporters to export paraquat, which notably is also a cation. We also observed strains with resistance; interestingly, these included ∆*aroP*, an aromatic amino acid permease knockout. Given the aromatic structure of paraquat, it is very plausible that one of the means of its entry into the cell is via this aromatic amino acid permease (Figure 7C). The most sensitive and resistant strains in paraquat are shown in Table 4.

### 2.5. Large-Scale Data Analysis

The generation of a dataset such as this one also allows the large-scale investigation of relationships between transporters. The heat map of correlations between AUC values for strains across growth conditions is shown in Figure 8A. We also extracted the *p* values for the correlation and found a peak near zero (Figure 8B), indicating likely true positives in the significant results. We found 35 correlations that met the conditions for significance at *p* < 0.05 after full Bonferroni correction [31] (with a corrected *p* value of 2.5 × 10^−7^). These are shown in Table 2. One possible application of this type of large-scale data analysis is in predicting substrates of unannotated transporters by their relationship to transporters with annotations (a “guilt by association” methodology) (Table 5). For instance, the growth parameters of ∆*sapB*, a putrescine exporter knockout, have a high correlation with the knockout of orphan transporter knockout ∆*ydjE* (Figure 8C), suggesting transport of compounds similar to putrescine or at least some overlap in substrate selectivity and function. 

## 3. Discussion

Identifying substrates and describing the structure activity relationship of bacterial transporters is a challenging task. Given the importance of transport systems in the function of most antibiotics, understanding the influx and efflux pathways of antimicrobial agents is highly relevant to mitigating problem of drug resistant bacteria. In this study, we have sought to gain insights into the potential influx and efflux pathways of a set of compounds with antimicrobial activity in *E. coli*. We have used an automated method to generate a dataset of growth curves for 447 transporter knockout strains against subinhibitory concentrations of 18 structurally diverse compounds with antibacterial activity. Our data showed results consistent with some previously reported pathways for antibiotic influx and efflux. We also report on numerous novel specific observations of transporter knockouts that reduce or enhance growth in the presence of antimicrobials as well as discuss potentially novel observations from analysis of the full dataset. All data are provided as a resource in the Appendix A. 

There were numerous novel transporter–compound interactions identified in this study, which suggest the need for follow-up experiments. We have previously developed a workflow for confirmatory work on the results that involves: (1) PCR to confirm the strain is correctly labelled, and (2) MIC determination to validate the observed result [20]. Work to automate these processes is ongoing and will be detailed in future publications. Once a specific transporter–compound interaction has been confirmed, the task of finding other, potentially higher affinity substrates for the transporter is feasible, as often the identification of an initial substrate is the most difficult step. Tools that can identify compounds “closest” to the antimicrobial in chemical space [32] will allow detailed investigation of the structure activity relationship of transporters. 

A limitation to the approach described in this study for transporter pathway identification is that growth is used as a proxy for transport with no direct measurement. While the simplest mechanism by which a transporter knockout changes growth in the presence of an antimicrobial is through the direct alteration of transport, there are other possible mechanisms. Gene knockouts are well-known to cause pleiotropic effects, with knockouts of a single gene often causing altered expression in many other genes that may affect transporter sensitivity [33]. Further validation of predicted effects can be achieved by use of the ‘ASKA’ overexpression collection [34], where, if the opposite effect is observed (e.g., resistance in an overexpression strain and sensitivity in a knockout strain), this could be seen as providing further evidence that the compound is indeed a substrate of the given transporter [19,35,36]. Given the large degree of redundancy in transporters (i.e., a given substrates may be transported by several transporters), more robust follow-up results could be achieved through investigation of double knockouts, as has recently been demonstrated in yeast [37]. 

There are studies that have investigated the growth of the full Keio collection of *E. coli* knockouts under different conditions (including numerous antibiotics) on solid media [27,38,39]. While the use of solid media and the application of advanced robotics allows a much higher throughput (with all of the above studies run in a 1536-well format), this does come with limitations. This high-density solid-media growth introduces a “neighbor effect”, where crowding of the bacterial colonies causes growth inhibition that needs to be corrected for [38]. Additionally, these studies report one metric or at most two metrics for growth, which potentially obscures the relevant information only available from the full growth curve. It should also be noted that while the authors of the above papers all commendably shared their data, in two out of the above three publications, the provided links for accessing data are no longer active [27,38], fitting with an observed trend of data accessibility decreasing in the years following publication [40]. 

There is a large degree of homology found between transporters of different pathogenic bacterial families. Given this, as well as the recent advances in prediction of protein structure from sequence [41,42], we envision that the dataset generated in this paper will be of use for understanding and predicting the interaction between antibiotics and membrane transporters in other clinically relevant bacterial species. 

## 4. Materials and Methods

Antibiotics were sourced from Sigma, apart from rifampicin, ceftriaxone and azithromycin, which were purchased from Tokyo Chemical Industries (Zwijndrecht, Belgium).

Plates were prepared by first inoculating deep-well plates with 1 mL Lucia Broth (LB) from glycerol stocks of the transporter library. The inoculated LB was grown overnight under agitation at 37 °C. Following overnight growth, the cultures were mixed with 1 mL 50% glycerol. CR1496c polystyrene plates (Enzyscreen, NL, Heemstede, The Netherlands) were prepared for growth assays, by dispensing a 3 µL droplet of the culture and glycerol mixture into the bottom of the well. These loaded plates were then stored at −20 °C for up to 3 weeks or −80 °C for longer storage. 

The growth assays were initiated by adding 297 µL LB, containing the appropriate dose of antibiotics to the pre-inoculated plates, and sealing with CR1396b Sandwich covers (Enzyscreen, NL, Heemstede). Growth was assessed using the Growth Profiler 960 (Enzyscreen, NL, Heemstede), which uses camera-based measurements to estimate growth rates simultaneously in up to 10 96-well plates. As our library encompassed 5 plates, this allowed us to run the full library in duplicate. The Growth Profiler 960 was set to 37 °C with 225 rpm shaking (recommended settings for *E. coli*), with pictures taken every 20 min.

Inoculation and media loading were performed using an Opentrons OT2 robot fitted with a 20 µL multichannel pipette and a 300 µL multichannel pipette. Scripts used in operation can be found at github.com/ljm176/TransporterScreening (11 August 2022). Growth plates were sterilized between uses by washing and UV in accordance with the instructions of the manufacturer.

*G*-values were obtained from the plate images using the manufacturer’s software. *G*-values were converted to *OD*600 values using the formula:$$OD_{600}=a*{\left(G_{value}-G_{Blank}\right)}^{b}$$

With the predetermined values *a* = 0.0158 and *b* = 0.9854, which were found by measurement of a standard curve in accordance with the instructions of the manufacturer.

Data analysis and generation of figures was performed in R (version 4.1.2). Growth rates were determined by using the R package Growthcurver [43]. Growthcurver fits growth curves to the equation:$$N_{t}=\frac{K}{1+\left(\frac{K-N0}{N0}\right)e^{-rt}}$$
where *N_t_* is total cell population, *K* is the carrying capacity, initial cell population is *N*0 and *r* is the intrinsic growth rate. For strains that showed growth in only a single replicate, the replicate without growth was filtered from analysis, as this was determined to be the result of missed inoculation during automated loading. 

Figures were generated using ggplot2 for R. The R script used in data analysis and figure generation is available at github.com/ljm176/TransporterScreening (11 August 2022). 

## Figures and Tables

**Figure 1 antibiotics-11-01129-f001:**
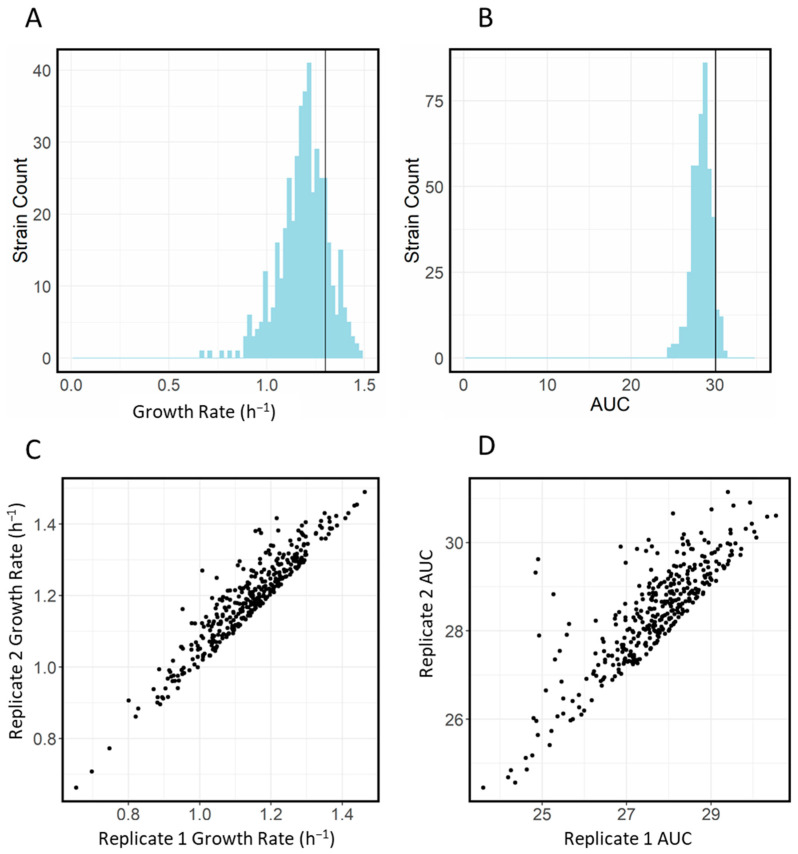
Histograms illustrating distributions of (**A**) growth rate and (**B**) area under the curve (AUC) for transporter library growth in LB in the absence of antibiotics. Scatter plots showing correlation between replicates for (**C**) growth rates and (**D**) AUC for transporter library growth in LB alone.

**Figure 2 antibiotics-11-01129-f002:**
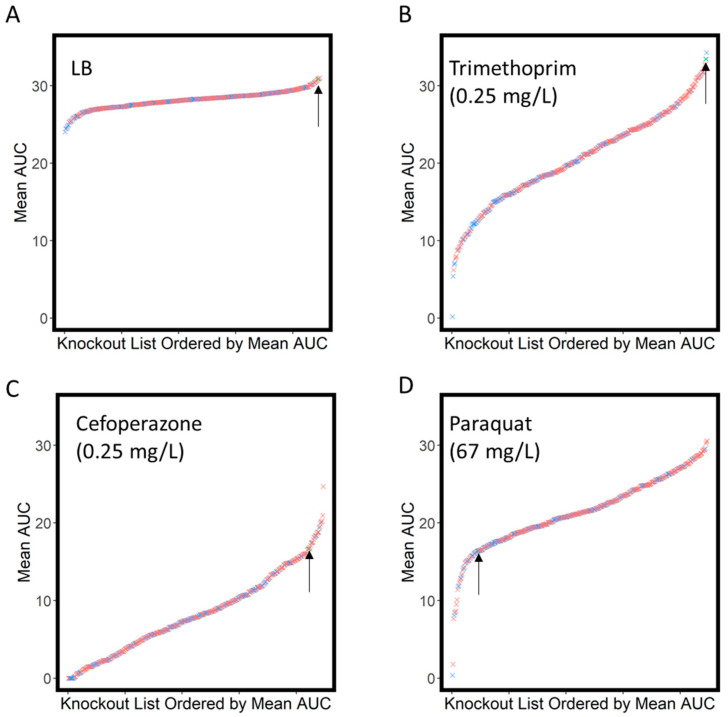
Transporter knockout library ordered by the mean area under the curve (AUC). Data shown for growth in LB (**A**) and subinhibitory concentrations of antibiotics/antimicrobials indicated (**B**–**D**). Y-genes are shown in blue, and annotated transporters are shown in red, while the arrow indicates the position of the WT strain.

**Figure 3 antibiotics-11-01129-f003:**
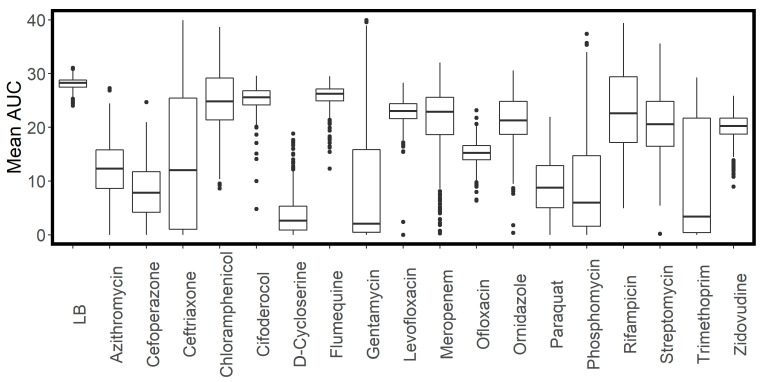
Box and whisker plot showing distribution of mean area under the curve (AUC) for growth of the transporter library in LB and antimicrobial compounds.

**Figure 4 antibiotics-11-01129-f004:**
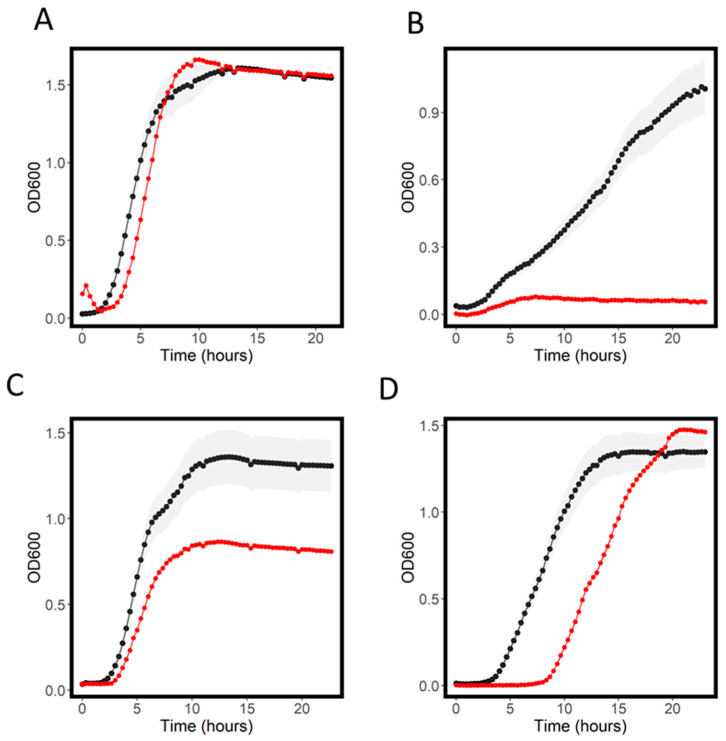
Growth curves for WT (black) and ∆*acrB* (red) in LB (**A**), azithromycin, (**B**) D-cycloserine (**C**) and chloramphenicol (**D**). Shaded areas represent the standard deviation for the WT growth curve.

**Figure 5 antibiotics-11-01129-f005:**
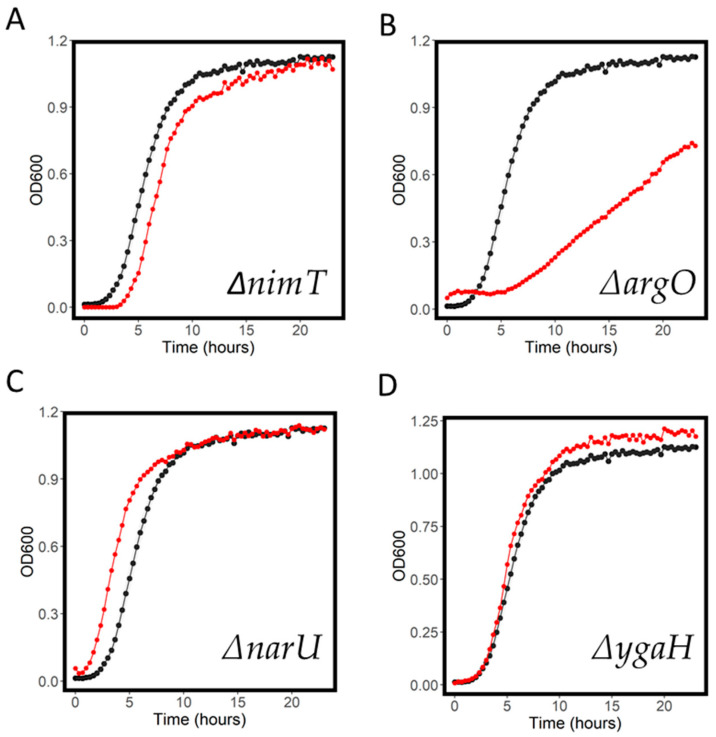
Growth curves for WT (black) and knockout strains (red) in 400 mg/L ornidazole. ∆*nimT* (**A**), ∆*argO* (**B**), ∆*narU* (**C**) and ∆*ygaH* (**D**) are shown. Shaded areas represent the standard deviation for the WT growth curve.

**Figure 6 antibiotics-11-01129-f006:**
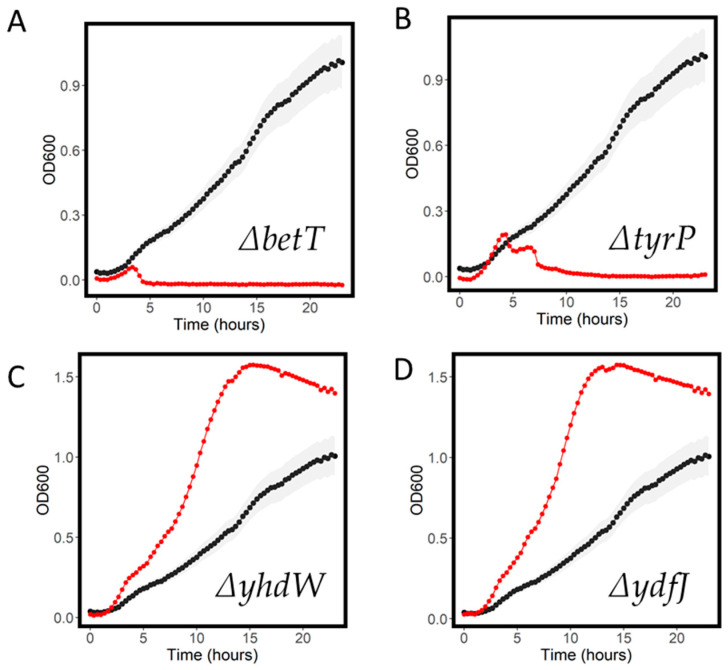
Growth curves for WT (black) and knockout strains (red) in 400 mg/L azithromycin. ∆*betT* (**A**), ∆*tyrP*(**B**), ∆*yhdW* (**C**) and ∆*ydfJ* (**D**) are shown. Shaded areas represent the standard deviation for the WT growth curve.

**Figure 7 antibiotics-11-01129-f007:**
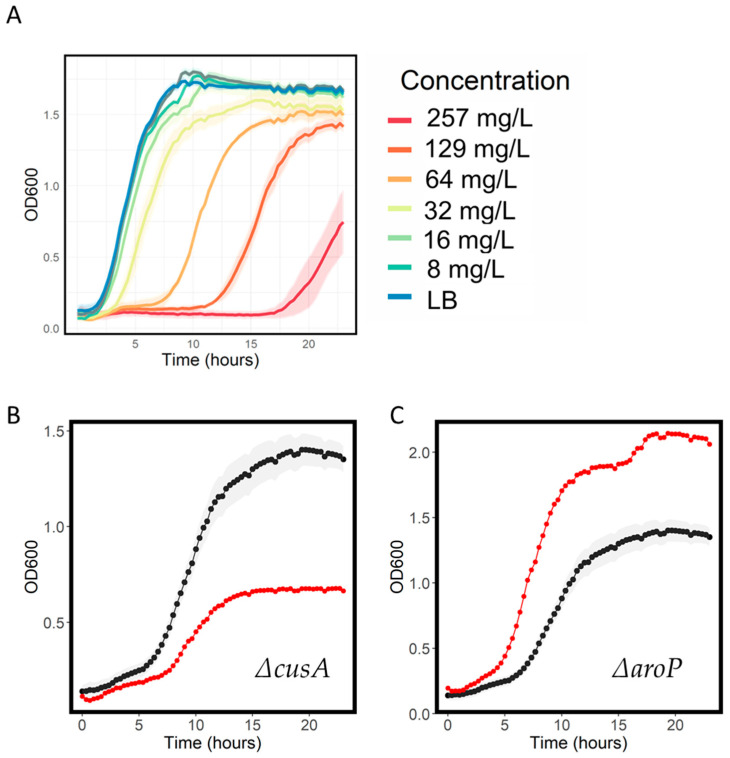
Growth features of paraquat. (**A**) Concentration inhibition data for growth of WT *E. coli* in the indicated concentration of paraquat. Shaded areas represent the standard deviation, n = 3. Growth curves for WT (black) and knockout strains (red) in 32 mg/L paraquat. ∆*cusB* (**B**) and ∆*aroP* (**C**) are shown.

**Figure 8 antibiotics-11-01129-f008:**
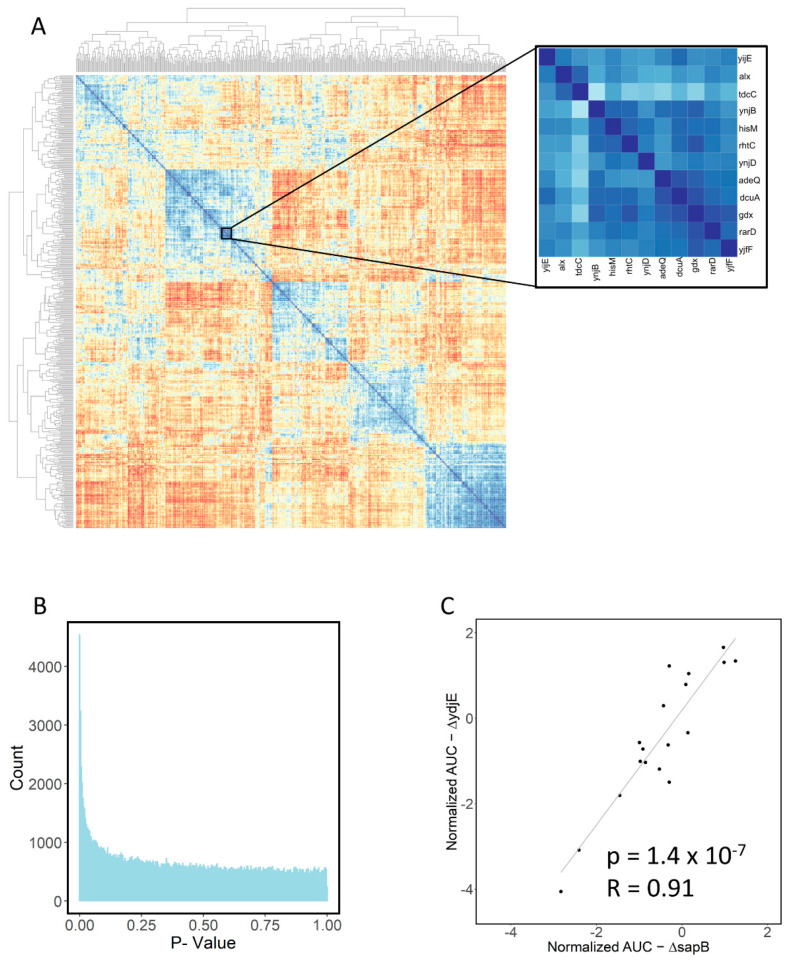
(**A**). Heatmap showing correlations between area under the curve values for transporters in all antibiotics. Zoom inset highlights a cluster of highly correlated transporter knockout strains. (**B**) Histogram of *p*-values for all correlations showing the peak around 0, indicating likely true positive results. (**C**) Scatter plot indicating high degree of correlation between ∆*sapB* and ∆*ydjE*.

**Table 1 antibiotics-11-01129-t001:** Antibiotics and antimicrobials used in this study, with minimum inhibitory concentration (MIC) in the wild-type (WT) strain, and the concentrations used for high-throughput screening experiments are shown.

Antibiotic	WT MIC (mg/L)	Screening Concentration (mg/L)
Azithromycin	30	7.5
Cefiderocol	1	0.25
Cefoperazone	0.5	0.25
Chloramphenicol	2.5	1.25
Ceftriaxone	2.5	1
D-Cycloserine	12.5	6.25
Flumequine	>1.5 *	1.5
Gentamycin	4	2
Levofloxacin	0.25	0.125
Meropenem	0.0625	0.03125
Ofloxaxin	0.125	0.0625
Ornidazole	800	400
Paraquat	>268 *	67
Phosphomycin	50	25
Rifampicin	25	12.5
Streptomycin	25	6.25
Trimethoprim	2.3	0.25
Zidovudine	>100 *	100

* For these compounds, growth was detected at the highest concentration tested, despite clear inhibition at lower concentrations.

**Table 2 antibiotics-11-01129-t002:** Growth parameters for the 15 most sensitive and most resistant strains in Ornidazole by normalized AUC. Normalized values show the value in ornidazole/value in LB for the given strain.

Strain	Max OD	Rate	AUC	Normalized Max OD	Normalized Rate	Normalized AUC
argO	0.74	0.35	6.36	0.29	0.45	0.23
ydjE	0.58	0.71	6.55	0.71	0.37	0.26
tolQ	0.60	1.05	7.98	1.16	0.38	0.29
glvB	0.75	0.57	9.78	0.54	0.44	0.34
sapB	0.64	0.73	9.05	0.74	0.41	0.35
yebQ	0.83	0.59	8.96	0.89	0.53	0.36
ddpF	0.76	0.80	11.43	0.64	0.44	0.37
uup	0.72	0.85	10.07	0.81	0.45	0.37
cysW	0.64	1.62	10.22	1.40	0.41	0.38
ydcS	0.65	0.54	9.25	0.70	0.39	0.38
ydeE	0.78	0.89	11.00	0.74	0.48	0.39
yicJ	0.81	0.84	11.78	0.64	0.49	0.39
guaB	0.69	1.08	11.22	1.18	0.42	0.40
gntU	0.87	1.04	12.13	0.83	0.52	0.40
ynfA	0.77	0.93	9.97	1.31	0.52	0.41
ybhF	1.15	0.95	19.30	0.96	0.66	0.65
arnE	1.14	1.03	17.97	1.14	0.70	0.66
sapC	1.11	0.74	17.12	0.77	0.67	0.66
cycA	1.17	1.07	19.65	0.83	0.69	0.66
narU	1.14	0.86	19.88	0.67	0.67	0.66
lacY	1.19	0.87	20.61	0.71	0.68	0.67
kup	1.16	0.90	18.67	0.68	0.70	0.67
mdtH	1.24	0.80	18.59	0.89	0.75	0.67
pitA	1.25	1.01	18.19	0.78	0.77	0.67
yadG	1.09	0.84	17.65	0.66	0.66	0.67
clcB	1.14	0.86	18.96	0.67	0.69	0.68
trkH	1.20	0.76	18.90	0.56	0.74	0.68
exbD	1.22	0.92	20.52	0.79	0.70	0.70
wt	1.04	1.10	21.76	0.77	0.58	0.70
dhaM	1.34	0.79	23.18	0.63	0.80	0.78

**Table 3 antibiotics-11-01129-t003:** Growth parameters for the 15 most sensitive and most resistant strains in ornidazole. Normalized values represent the value in ornidazole/value in LB for the given strain and values for the maximum OD, rate and area under the curve are shown.

Strain	MaxOD	Rate	AUC	Normalized MaxOD	Normalized Rate	Normalized AUC
betT	0.06	0.02	0.30	0.02	0.04	0.01
glvB	0.11	0.03	0.32	0.03	0.07	0.01
glnP	0.11	0.02	0.33	0.02	0.06	0.01
citT	0.06	0.04	0.37	0.04	0.04	0.01
tyrP	0.24	0.01	0.56	0.01	0.15	0.02
corA	0.14	0.03	0.62	0.02	0.08	0.02
tolR	0.11	1.69	0.67	1.46	0.07	0.02
mdtN	0.67	0.50	0.70	0.40	0.40	0.02
ydjN	0.34	0.44	0.76	0.38	0.20	0.03
yibH	0.20	0.01	0.77	0.00	0.12	0.03
torT	0.18	0.04	0.80	0.03	0.11	0.03
acrB	0.09	0.66	1.35	0.54	0.05	0.05
kdgT	2.16	0.92	1.48	0.75	1.33	0.05
tolQ	0.74	1.04	1.76	1.16	0.47	0.06
ydcS	0.31	0.28	1.61	0.36	0.18	0.07
ddpD	1.51	0.39	19.71	0.30	0.92	0.71
ydiN	1.53	0.44	20.19	0.36	0.96	0.71
ydjX	1.58	0.43	20.37	0.35	0.96	0.72
yddB	1.59	0.40	19.66	0.28	1.00	0.72
atpB	1.57	0.44	21.09	0.35	0.96	0.73
yccA	1.54	0.43	19.30	0.37	0.99	0.73
ygaZ	1.50	0.58	18.50	0.56	1.09	0.73
cmtA	1.54	0.41	20.84	0.36	0.95	0.74
kgtP	1.52	0.47	21.29	0.38	0.88	0.74
pitA	1.51	0.46	20.38	0.36	0.93	0.75
ydfJ	1.58	0.55	23.16	0.41	0.93	0.76
yhdW	1.58	0.48	22.07	0.34	0.95	0.76
yraQ	2.07	0.81	24.46	0.75	1.23	0.87
yiaM	2.51	0.51	26.88	0.42	1.50	0.90
argO	1.92	0.94	27.30	0.78	1.17	0.97

**Table 4 antibiotics-11-01129-t004:** Growth parameters for the 15 most sensitive and most resistant strains in paraquat. Normalized values represent the value in paraquat/value in LB for the given strain and values for the maximum OD, rate and area under the curve are shown.

Strain	MaxOD	Rate	AUC	Normalized MaxOD	Normalized Rate	Normalized AUC
ydcZ	0.11	1.55	0.39	1.35	0.06	0.01
potH	0.74	0.57	1.79	0.50	0.44	0.06
fieF	1.46	1.20	7.66	0.92	0.85	0.26
yihO	1.35	1.07	8.11	0.88	0.79	0.29
kdgT	2.22	0.49	8.51	0.40	1.36	0.29
cusA	0.69	0.49	8.66	0.50	0.41	0.29
yfdV	0.78	0.60	8.42	0.64	0.47	0.30
uhpC	2.15	0.55	9.50	0.48	1.27	0.32
uacT	2.40	0.50	10.09	0.40	1.42	0.33
atoS	1.47	0.26	11.42	0.25	0.84	0.38
yihN	1.11	0.64	11.91	0.55	0.65	0.41
ygjI	0.97	0.53	11.79	0.41	0.58	0.42
ygaH	1.41	1.01	13.21	0.85	0.82	0.42
mdtN	0.96	0.46	12.28	0.37	0.57	0.43
yidK	1.03	1.04	12.84	0.75	0.60	0.43
aroP	2.15	0.60	28.71	0.51	1.30	1.03
gsiC	2.21	0.68	28.48	0.64	1.37	1.03
yaaU	2.15	0.55	28.40	0.47	1.33	1.03
copA	2.15	0.57	28.30	0.50	1.30	1.03
araJ	2.16	0.64	28.94	0.52	1.32	1.03
dinF	2.13	0.60	29.45	0.47	1.27	1.04
codB	2.14	0.58	29.43	0.47	1.27	1.04
sbp	2.01	0.67	27.00	0.53	1.27	1.04
amtB	2.29	0.60	29.30	0.53	1.41	1.05
kdpA	2.34	0.60	29.41	0.50	1.42	1.05
yfeO	2.01	0.64	28.17	0.49	1.22	1.06
uidB	2.25	0.55	30.25	0.44	1.36	1.06
caiT	2.34	0.57	30.46	0.45	1.41	1.06
acrB	2.15	0.52	29.25	0.43	1.28	1.07
focA	2.26	0.61	30.58	0.52	1.37	1.09

**Table 5 antibiotics-11-01129-t005:** Transporters that are highly correlated.

Transporter 1	Transporter 2	Correlation	*p* Value
yihP	yhdX	0.91	2.46 × 10^−7^
ydjK	mdtB	0.91	2.36 × 10^−7^
uup	gsiC	0.91	2.36 × 10^−7^
yicL	yhdX	0.91	2.26 × 10^−7^
btuC	araJ	0.91	2.22 × 10^−7^
mdtG	copA	0.91	2.04 × 10^−7^
rarD	adeQ	0.91	2.02 × 10^−7^
mscK	ccmC	0.91	1.98 × 10^−7^
cynX	btuC	0.91	1.72 × 10^−7^
ybbA	cynX	0.91	1.65 × 10^−7^
mdfA	chaA	0.91	1.49 × 10^−7^
yjfF	gdx	0.91	1.24 × 10^−7^
ynjB	gdx	0.91	1.18 × 10^−7^
nhaB	ddpB	0.91	1.16 × 10^−7^
ydjE	sapB	0.91	1.14 × 10^−7^
yphD	gudP	0.91	1.07 × 10^−7^
ybaT	btuC	0.92	9.3 × 10^−8^
yjfF	adeQ	0.92	8.62 × 10^−8^
ypjA	fryA	0.92	5.86 × 10^−8^
rarD	gdx	0.92	5.63 × 10^−8^
malX	amtB	0.92	5 × 10^−8^
araJ	acrB	0.92	4.29 × 10^−8^
gsiC	fetA	0.93	3.67 × 10^−8^
yaaU	sapA	0.93	3.54 × 10^−8^
uidB	mdtG	0.93	3.53 × 10^−8^
yphD	ascF	0.93	2.77 × 10^−8^
yegT	frwC	0.93	2.61 × 10^−8^
ytfT	kup	0.93	1.47 × 10^−8^
copA	ccmC	0.93	1.46 × 10^−8^
yddB	ddpD	0.94	7.5 × 10^−9^
ybbA	mscK	0.95	2.46 × 10^−9^
mscK	fetA	0.95	8.93 × 10^−10^
cynX	araJ	0.96	3.92 × 10^−10^
gdx	adeQ	0.97	5.67 × 10^−11^
yihP	yicL	0.97	7.51 × 10^−12^

## Data Availability

Datasets are available in the Appendix A for this paper. The code used in analysis is available at github.com/ljm176/TransporterScreening.

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
