# Peer review of "Analysis of a Library of Escherichia coli Transporter Knockout Strains to Identify Transport Pathways of Antibiotics"

_antibiotics, 2022, doi:10.3390/antibiotics11081129_

Round 1
Reviewer 1 Report
This study is proposed to identify substrates and describe the structure-activity relationship of bacterial transporters, improving the understanding of the influx and efflux pathways of antimicrobial agents. In this study, they have used an automated method to generate a dataset of growth curves for 447 transporter knockout strains against sub-inhibitory concentrations of 18 structurally diverse compounds with antibacterial activity. These data showed results consistent with pathways for antibiotic influx and efflux and reported numerous knockout strains that showed resistant or sensitive phenotypes to specific antimicrobials. This study is very interesting and appropriate for journal audiences—some revisions to follow.
1) Improve the figures by adding units of measure to the axes, improving and simplifying the captions of the images and tables in the manuscript. The reader should understand more simply and correctly what is shown in the figures and tables.
2) Line 78: reports in more detail and precisely the detection of the bacterial growth observed.
3) Could other experiments and improvements in the methods used to be carried out to better support the results obtained?
Author Response
We thank the reviewer for their comments. Point by point responses are below.
1) Improve the figures by , improving and simplifying the captions of the images and tables in the manuscript. The reader should understand more simply and correctly what is shown in the figures and tables.
Note that AUC is a normalised quantity and is thus unitless. Units have been added to other axes.
2) Line 78: reports in more detail and precisely the detection of the bacterial growth observed.
This has been added.
3) Could other experiments and improvements in the methods used to be carried out to better support the results obtained?
Further experiments that would validate the results are discussed in the second and third paragraph in the discussion. We have also included more clarification that this study is exploratory and our future work will focus on targeted follow up validation experiments.
Reviewer 2 Report
This study used a large E.coli single transporter gene knockout library (Keio collection) and growth curve analysis in the presence of (sub-)inhibitory concentrations of various drugs to screen for possibly new transporter-substrate associations. Growth curve parameter estimation was done with the help of a Growth Profiler 960 instrument with image analysis hard- and software. The endpoint of analysis was altered growth compared to the wild-type strain in the presence of drugs (usually half the MIC). The workflow has previously described (in a study searching for melatonin efflux transporters). The library included knockouts of previously unannotated transporters or y-genes but also e.g. the transcriptional regulator SoxR of E. coli. The list of tested compounds was said to include 18 structurally diverse compounds with antibacterial activity, but it included both ofloxacin and levofloxacin, ceftriaxone and cefoperazone, and gentamicin and streptomycin. Also it included D-cycloserine (instead of e.g. linezolid which is a known substrate of
Comments and suggestions:
General:
- The rationale for the selection of the compounds is not given. Also, the exact (and perhaps optimal [for an automated method] definition of relevant endpoints (using differential growth expressed by growth rate or growth AUC differences) has not been described and discussed.
- There is considerable plasticity and redundancy among the transporters limiting single gene knockout experiments. This should be commented.
- The authors report only very selective results: azithromycin, D-cycloserine, and chloramphenicol for ∆acrB, ornidazole for several knockouts, and paraquat for ∆cusB and ∆aroP, respectively. It is not stated why those results were selected (based on an endpoint algorithm ? arbitrarily ?)
- It woud be very useful to briefly but explicitly include an analysis of control (known) substrate-transporter pairs, e.g. macrolide-macB, ofloxacin and levofoxacin-acrB and acrF, fosfomycin-cusA and fosfomycin-uhpT/-glpT, gentamicin-acrD etc.
Minor comments
- The differences between wt and knockouts (normalized max OD, normalized rate and normalized AUC) do not appear to identify distinct outliers (susceptible and resistant): again, what are the definitions used to identify relevant phenotypes for further study ?
- The provision of the excel table (supplementary material) is appreciated
- The authors state that „some strains also showed an ornidazole resistant phenotype“ (page 7) which is suggested by the growth curves in figure 5 but not clear from table 2 looking at the AUCs.
- The legend for table 3 is missing
- Figure 6: legend incorrectly states ornidazole (instead of azithromycin) ?
- Despite < 1/2 MIC for ceftriaxone growth was very poor in many strains. Any explanation ?
- Any reason why the full MIC was tested in the case of cefiderocol ?
- Pages 12/13: it is not stated what values entered into the correlation analysis (not in the methods section, not in the text, not in the figure)
Author Response
We thank the reviewer for the insightful comments, and also for a very thorough reading that found some small errors we had missed during preparation. We have responded point by point to the comments below.
Comments and suggestions:
General:
- The rationale for the selection of the compounds is not given. Also, the exact (and perhaps optimal [for an automated method] definition of relevant endpoints (using differential growth expressed by growth rate or growth AUC differences) has not been described and discussed.
Some more explanation as to our rationale for not including explicit enpoints has been included, and a paragraph with a brief explanation as to our selection criteria has been added to the start of the introduction.
- There is considerable plasticity and redundancy among the transporters limiting single gene knockout experiments. This should be commented.
Commentary on this has been added to the discussion (lines 296-297)
- The authors report only very selective results: azithromycin, D-cycloserine, and chloramphenicol for ∆acrB, ornidazole for several knockouts, and paraquat for ∆cusB and ∆aroP, respectively. It is not stated why those results were selected (based on an endpoint algorithm ? arbitrarily ?)
Our strategy has been clarified in the text (lines 127-139).
- It woud be very useful to briefly but explicitly include an analysis of control (known) substrate-transporter pairs, e.g. macrolide-macB, ofloxacin and levofoxacin-acrB and acrF, fosfomycin-cusA and fosfomycin-uhpT/-glpT, gentamicin-acrD etc.
This is a good suggestion, and we have added some explicit discussion of previously reported transporter/substrate pairs in lines 146 to 155.
Minor comments
- The differences between wt and knockouts (normalized max OD, normalized rate and normalized AUC) do not appear to identify distinct outliers (susceptible and resistant): again, what are the definitions used to identify relevant phenotypes for further study ?
This has been clarified in lines 127-139.
- The provision of the excel table (supplementary material) is appreciated
- The authors state that „some strains also showed an ornidazole resistant phenotype“ (page 7) which is suggested by the growth curves in figure 5 but not clear from table 2 looking at the AUCs.
The strains highlighted did not make the top 15 strains (which we arbitrarily selected as a cutoff) but we have clarified our reasons for discussing these in particular, which had clearly altered phenotypes compared to WT.
- The legend for table 3 is missing
This has been added.
- Figure 6: legend incorrectly states ornidazole (instead of azithromycin) ?
This error has been rectified.
- Despite < 1/2 MIC for ceftriaxone growth was very poor in many strains. Any explanation ?
At concentrations close to the MIC then small variations in concentration between the MIC determination experiment and the screening experiment may result in large differences in growth. This is our understanding of what happened, however since we required only a concentration that allowed us to see growth differences between strains we don’t feel this is a major problem.
- Any reason why the full MIC was tested in the case of cefiderocol ?
This was well spotted by the reviewer (if referring to the fact that the same number was entered for screening concentration and MIC in table 1). This was an error in filling in the table and has been corrected to indicate the actual MIC and screening concentration.
- Pages 12/13: it is not stated what values entered into the correlation analysis (not in the methods section, not in the text, not in the figure)
This has been added.
Reviewer 3 Report
Dear author,
Excellent scientific work. The following changes must be made. Supplement the abstract with more information about the conducted research and its significance. Review the structure of the article, I think the materials and methods should come before the results
Author Response
We thank the reviewer for the review and suggestions. We have expanded the abstract as requested. As we are following a structure provided by the MDPI Antibiotics template we must leave it in the suggested structure (unless editors suggest otherwise).
Reviewer 4 Report
In the world today, increased antimicrobial resistance (AMR) is a serious problem. Multidrug-resistant strains (MDR) emerge among bacteria, which poses a great threat to the health and life of patients. E.coli are one of the species belonging to the Enterobacteriaceae family, responsible for nosocomial and environmental infections.Overall, this is a concise manuscript mainly focusing on the whole genome sequencing data analysis.
Author Response
We thank the reviewer for the positive feedback.
Round 2
Reviewer 1 Report
The suggestions were carried out, the manuscript was improved. The section on materials and methods could be enriched, being very poor in content, compared to the data reported in the results, as the last suggestion.
Author Response
We have added some more detail to sections of the materials and methods.
Reviewer 2 Report
The comments/suggestions have been addressed.
Author Response
We thank the reviewer for their input.